# NMDA and AMPA Receptors at Synapses: Novel Targets for Tau and α-Synuclein Proteinopathies

**DOI:** 10.3390/biomedicines10071550

**Published:** 2022-06-29

**Authors:** Maria Italia, Elena Ferrari, Monica Diluca, Fabrizio Gardoni

**Affiliations:** Department of Pharmacological and Biomolecular Sciences, University of Milan, 20133 Milan, Italy; maria.italia@unimi.it (M.I.); elena.ferrari2@unimi.it (E.F.); monica.diluca@unimi.it (M.D.)

**Keywords:** synaptic dysfunction, NMDA receptors, AMPA receptors, tau, α-synuclein, dendritic spines

## Abstract

A prominent feature of neurodegenerative diseases is synaptic dysfunction and spine loss as early signs of neurodegeneration. In this context, accumulation of misfolded proteins has been identified as one of the most common causes driving synaptic toxicity at excitatory glutamatergic synapses. In particular, a great effort has been placed on dissecting the interplay between the toxic deposition of misfolded proteins and synaptic defects, looking for a possible causal relationship between them. Several studies have demonstrated that misfolded proteins could directly exert negative effects on synaptic compartments, altering either the function or the composition of pre- and post-synaptic receptors. In this review, we focused on the physiopathological role of tau and α-synuclein at the level of postsynaptic glutamate receptors. Tau is a microtubule-associated protein mainly expressed by central nervous system neurons where it exerts several physiological functions. In some cases, it undergoes aberrant post-translational modifications, including hyperphosphorylation, leading to loss of function and toxic aggregate formation. Similarly, aggregated species of the presynaptic protein α-synuclein play a key role in synucleinopathies, a group of neurological conditions that includes Parkinson’s disease. Here, we discussed how tau and α-synuclein target the postsynaptic compartment of excitatory synapses and, specifically, AMPA- and NMDA-type glutamate receptors. Notably, recent studies have reported their direct functional interactions with these receptors, which in turn could contribute to the impaired glutamatergic transmission observed in many neurodegenerative diseases.

## 1. Introduction

Protein misfolding, accumulation and the formation of toxic aggregates and fibrils are common events in neurodegenerative disorders, also referred to as proteinopathies. Aberrant levels, post-translational modifications (PTMs), mislocalisation, and aggregation of tau and α-synuclein have been widely described in Alzheimer’s disease (AD) and related tauopathies and Parkinson’s disease (PD) and related synucleinopathies, respectively. Tau and α-synuclein are highly expressed in the central nervous system and their toxic forms can lead to detrimental effects in neurons, including synaptic dysfunction during early disease stages, affecting several pathways. However, the molecular mechanisms by which tau and α-synuclein aggregates impair synaptic transmission and plasticity have not been fully elucidated. Here, we focused on recent evidence addressing a specific role for ionotropic glutamate receptors, namely α-amino-3-hydroxy-5-methyl-4-isoxazolepropionic acid receptors (AMPARs) and N-methyl-D-aspartate receptors (NMDARs), in modulating postsynaptic toxicity of both tau and α-synuclein.

## 2. Tau

### 2.1. Physiological Function of Synaptic Tau

For decades, the physiological function of microtubule-associated protein tau has been linked to microtubule stabilisation and axonal elongation in the somato-axonal compartment [1]. The presence of tau in the dendritic compartment was considered an index of toxicity and was often associated with neurodegeneration.

However, the discovery of tau in both pre- and post-synaptic structures of healthy neurons has led to the hypothesis that tau could also exert a physiological function in dendritic spines [2]. In this regard, it has been demonstrated that tau not only migrates to the spines, but is also locally translated [3]. 

As the physiological presence of tau in dendritic spines has been proven, it is now of interest to understand what its synaptic function is. The aim of the following section discusses this aspect with a specific focus on AMPAR and NMDAR crosstalk with tau.

#### 2.1.1. Tau as a Mediator of Synaptic Plasticity

Synaptic plasticity encompasses a series of mechanisms by which neurons modify the strength of their connections in an activity-dependent manner. Long-lasting forms of synaptic plasticity, termed long-term potentiation (LTP) and long-term depression (LTD), are required for processes such as learning and memory formation.

Kimura and colleagues [4] demonstrated that tau is required for NMDAR-dependent LTD in the hippocampus. First, they adopted a genetic approach using tau KO mice. They showed that while LTD was readily induced in adult MAPT+/+, it was completely absent in MAPT+/− and MAPT−/− mice. No differences in LTP were observed comparing the three genotypes, indicating a specific impairment of LTD in the absence of tau in vivo. Additionally, Kimura et al. identified the same impairment in brain slices prepared from young MAPT−/− mice, suggesting that the LTD impairment in tau KO mice was present early in development, and was not the result of ageing. To exclude other confounding factors due to developmental complications resulting from the absence of tau throughout the life of the animals, they also investigated the effects of acute suppression of tau through specific shRNA administered to neurons from organotypic cultured hippocampal slices. Again, the only impairment that they found was in LTD. As further proof, when the silenced tau was replaced with human tau, the neuronal ability to exhibit LTD was completely restored.

Subsequently [5], Regan and colleagues looked for a behavioural counterpart of the LTD deficit identified in tau KO mice. Previous studies did not report significant differences in cognition in tau KO animals when compared to WT mice [6]. However, Regan et al. focused on a form of cognition strictly related to LTD: hippocampus-dependent spatial reversal learning. In their study, tau KO mice showed a pronounced impairment in ability to forget one location and learn another within a specific context, an aspect which is strictly related to LTD expression [7,8].

After establishing the necessity for the presence of tau in NMDAR-dependent LTD, they endeavoured to identify the mechanism through which tau exerts its function in this form of synaptic plasticity. PICK1-mediated AMPAR endocytosis from the post-synaptic membrane is a crucial event that supports the weakening of synapses during LTD. Regan et al. observed that in tau KO mice the physiological interaction between GluA2 and PICK1 triggered by LTD induction was impaired [5]. This result suggested that the function of tau in LTD could be related to AMPAR internalisation through PICK1. This supports the findings of other studies, which proposed a role for tau in intra-dendritic trafficking of AMPAR in both physiological and pathological conditions [9,10]. 

Another mechanism through which tau may affect NMDAR-dependent LTD expression could be related to NMDAR targeting. Ittner and colleagues [11] demonstrated the involvement of tau in post-synaptic targeting of Fyn. In their model, they proposed that the Src tyrosine kinase Fyn localises to the post-synapse in a tau-dependent manner, where it associates with PSD95; this association promotes Fyn kinase activity on the GluN2B subunit (GluN2B). This phosphorylation promoted the interaction of NMDARs with PSD95; therefore, tau presence in the spine ultimately increased the stability of NMDARs within the postsynaptic density complex (see Figure 1, panel A). 

In the above studies, no deficits in LTP induction were observed upon knocking out or silencing tau. However, Sydow and colleagues [13] showed that LTP induction was inhibited when tau was over-expressed. This effect could still be related to the role of tau in LTD; an excessive presence and activation of tau may induce chronic LTD that could manifest as impaired LTP.

Overall, these studies supported a physiological role for tau in the synaptic compartment and, specifically, in synaptic plasticity, as well as in NMDAR and AMPAR synaptic localisation and trafficking regulation.

#### 2.1.2. A Physiological Role in the Spine for Tau Phosphorylation

Tau phosphorylation has always had negative connotations. However, emerging evidence has demonstrated that tau phosphorylation could regulate tau in a physiological manner; not only do different signatures of phosphorylation characterise tau species in different subcellular domains [14], but different signatures of tau evoke different forms of synaptic activity (see Figure 2).

In this regard, Mondragon-Rodriguez and colleagues [12] showed that the phosphorylation of tau controlled its interaction with the PSD-Fyn-NMDAR complex, therefore influencing synaptic activity. They determined that the activation of NMDA receptors led to phosphorylation of different sites in tau (Ser199, Ser202, Thr231, Ser235, Ser396, Ser404), which in turn modulated the interaction of tau with its partners. Specifically, phosphorylation of tau strengthened the tau-Fyn interaction and weakened the tau–PSD95 interaction. As a result, NMDAR-dependent changes in the phosphorylation status of synaptic tau could lead to a transient increase in synaptic Fyn and, consequently, a transient increase in NMDA receptor activation before tau–Fyn leaves the PSD95–NMDA receptor complex (see Figure 1, panel B). 

Similarly, it was suggested that glycogen synthase kinase-3 (GSK3) β-mediated phosphorylation of tau in the PHF epitope (rather than tau itself) is required for LTD expression. Kimura and colleagues [4] showed that LTD induction led to an increase in tau PHF epitope (Ser396 and Ser404) phosphorylation. Furthermore, Regan et al. [5] demonstrated that LTD could not be induced in the presence of a mutant form of tau resistant to phosphorylation in Ser396. This indicated that the observed increase in tau phosphorylation upon LTD induction is not simply a by-product of enhanced activity of LTD-signalling molecules, such as GSK3β, but has a role in LTD induction. The sole inhibiting action of GSK3β on tau was sufficient to impair the PICK1–GluA2 interaction upon LTD induction. Similarly, other studies had already reported that GSK3β was required for LTD [15] and its activation necessary for AMPAR internalisation [16,17].

Draffin and colleagues [18] proposed a controversial but thought-provoking theory. GSK3 exists in two isoforms, GSK3β and GSK3α. These isoforms share a similar structure, especially in the catalytic domain, which is why one isoform can be easily confused with the other (namely, GSK3β with GSK3α). They demonstrated that GSK3α, but not β (as indicated by other studies, some of which are reported above), is required for LTD expression. Additionally, they showed that “the presence of tau is required for the regulated anchoring of GSK3α during LTD […] suggesting that tau is an intrinsic component of the mechanism by which GSK3α induces synaptic depression”. This suggested that tau exerts its function in LTD both as a signalling molecule downstream GSK3 activation and as an upstream factor by mediating GSK3α engagement in the spine in response to NMDAR activation. On one hand, GSK3 requires tau to be recruited to the spine upon LTD induction; on the other, the GSK3-mediated phosphorylated form of tau is essential for LTD expression.

Overall, the studies above demonstrated that tau and its phosphorylated form play a crucial role in the post-synaptic compartment. This showed it is possible that the hyper-phosphorylation and mislocalisation of tau, observed in many different neurodegenerative diseases, could mediate toxic effects not only because of a gain of function (as traditionally described) but also because of a loss of its physiological function. In the next section, we describe in vitro and in vivo/ex vivo studies in which a dysregulation of synaptic function of tau was observed.

### 2.2. The Pathological Counterpart of Synaptic Tau

Beyond the physiological role exerted by tau in synapse functioning, there has been a growing interest in the question of how the pathological hyper-phosphorylation of tau has a detrimental impact on synapse functionality. 

#### 2.2.1. In Vitro Models

Recently, several models for the pathological role of tau in synapse functionality have been proposed through in vitro studies in cultured hippocampal neurons. In this section, we discuss these models, trying to subsume them into a single paradigm, and the implications they have on our current knowledge of the course of tauopathy disease. Importantly, even though neuronal cultures represent a simplification of the complexity of the brain and of the effects that misfolded tau has on brain circuits, these models give us insights into the mechanisms that likely precede the macroscopic effects (i.e., synapse loss and neuronal death) that are observed for in vivo models of tauopathy. 

All these models stem from a fundamental observation: when tau aberrantly mislocalises to dendritic spines, an impairment in glutamatergic synaptic transmission becomes evident. In presence of tau mislocalisation, several studies reported a reduction in the frequency and amplitude of miniature excitatory postsynaptic currents (mEPSCs) and in the synaptic expression of AMPAR and NMDAR [9,19,20]. 

From this premise, great effort has been placed in trying to understand whether a causal relationship exists between tau mislocalisation and glutamatergic synaptic transmission and, if verified, what the mechanisms are through which this happens.

In this regard, Braun and colleagues [21] showed that the absence of tau (i.e., hippocampal neurons derived from MAPT−/− animals) prevented all post-synaptic deficits that they observed in their model of chronic traumatic encephalopathy when tau was present. Similarly, other studies demonstrated that preventing the aberrant delocalisation of tau in spines resulted in no detection of deficits in glutamatergic transmission [19,20]. 

In this regard, the conditio sine qua non for tau mislocalisation is its aberrant phosphorylation. Indeed, both the inhibition of the major kinases of tau (cyclin-dependent kinase 5 (CDK5) and GSK3β) and the introduction of phosphorylation-resistant residues in the tau sequence, prevented its aberrant localisation in dendritic spines. It is likely that tau–microtubule interactions are disturbed by the modifications to the phosphorylation state of tau which, in turn, dissociates from microtubules to localise in spines [22,23].

Together, these results indicate that an alteration in tau state is sufficient to determine functional glutamatergic deficits and that it is tau hyper-phosphorylation and subsequent delocalisation in spines that triggers the toxic effects on synapses.

Focusing on hyper-phosphorylation, Teravskis and colleagues [20] tried to determine whether the phosphorylation of specific domains of tau could regulate its mislocalisation and the subsequent detrimental effects on the post-synaptic compartment. In their work, they conclusively demonstrated that “Phosphorylation in two discrete tau domains regulates a stepwise process leading to postsynaptic dysfunction”. Specifically, phosphorylation of Ser396 and Ser404 in the C-terminal domain promotes tau mislocalisation in spines, whereas the additional phosphorylation of specific residues in the proline-rich N-terminal domain (Ser202, Thr205, Thr212, Thr217 and Thr231) is required for tau to exert toxic effects on glutamatergic neurotransmission.

It must be noted that the aberrant phosphorylation of tau results from different events. Liao’s research group [9,19] demonstrated that both the presence of genetic mutations in the tau gene (namely, the well-characterised P301L tau mutation) and the presence of Aβ oligomers, support tau hyper-phosphorylation. More importantly, the presence of Aβ and the P301L mutation led to synaptic deficits only when tau could be phosphorylated (and not in presence of amino acid substitutions that make tau resistant to phosphorylation). Additionally, other factors, such as mechanical injury, promote tau hyper-phosphorylation and mislocalisation in spines [21]. These observations suggested that a wide range of toxic events may potentially converge to a common pathological pathway in which tau ultimately promotes synaptic deficits. 

Having demonstrated the existence of a causal relationship between tau mislocalisation and synaptic deficits, Miller and colleagues [19] investigated the mechanism through which tau negatively affects synaptic functionality. They observed that calcineurin plays an important role in this process. Calcineurin is a serine/threonine phosphatase which also mediates AMPAR endocytosis during LTD by acting on residue S845 of the GluA1 subunit. Moreover, calcineurin interacts with a segment of the proline-rich domain of tau. Therefore, Miller et al. hypothesised that the hyper-phosphorylation of tau could positively affect this interaction by enhancing its binding affinity with calcineurin and, consequently, promoting calcineurin accumulation in spines. In their model, the resulting enhanced de-phosphorylation activity of calcineurin on GluA1 promoted AMPAR internalisation and subsequent synapse depotentiation. Most relevant to our discussion, they showed that calcineurin inhibition rescued the disruption of AMPAR clustering caused by tau. In this way, they demonstrated that calcineurin causes, at least in part, tau-triggered synaptic deficits, therefore elucidating the mechanism through which tau mislocalisation affects glutamatergic signalling. A role for calcineurin in Aβ driven synaptic deficits has already been proposed; Wu and colleagues demonstrated that Aβ mediates neurodegeneration, at least in part, by activation of calcineurin and, subsequently, the nuclear factor of activated T cells (NFAT)-mediated downstream cascades [24].

From these studies, a central role for tau hyper-phosphorylation had been hypothesised and demonstrated. However, there could be factors other than tau hyper-phosphorylation contributing to tau mislocalisation. Caspase-2 cleavage of tau at Asp314 seemed to play a crucial role in this process. Indeed, Zhao and colleagues [25] showed that production of this cleaved form of tau (called Δtau314) promoted the mislocalisation of not only Δtau314 but also full-length tau into synapses. Moreover, they demonstrated that this event, in turn, supported a reduction in AMPAR trafficking and anchoring in dendritic spines, and eventually led to impairment in synaptic transmission. Perhaps most importantly, they showed that this specific cleavage was a required step for P301L tau to mislocalise in spines and cause synaptic deficits. This was demonstrated through the expression of tau mutants resistant to caspase-2 cleavage, which prevented tau from infiltrating spines, dislocating glutamate receptors, and impairing synaptic function in cultured neurons.

These data suggested that more than one bottleneck exists for avoiding tau mislocalisation, and that multiple PTMs support tau mislocalisation and it is their additive effect that ultimately causes tau to exert toxicity on the post-synaptic compartment.

In conclusion, it is evident that tau begins to negatively affect synaptic functionality (and, specifically, glutamatergic signalling) in an extremely early phase in the progression of tauopathies. In all the studies discussed here, no spine loss was detected; it is plausible that tau mislocalisation in spines and the subsequent alteration in AMPAR signalling are the very first markers of tau-mediated neurodegeneration.

Up to this point, we have described the pathological effects of the soluble form of tau in the synapse. Contrastingly, Shrivastava and colleagues [26] adopted an in vitro model (i.e., primary hippocampal neuronal cultures) to investigate the effects of tau fibrils on neuron functionality. Indeed, it has been documented that high molecular weight assemblies of Tau have prion-like properties; once released by the affected neuronal cells, they are taken up by healthy cells and amplified by promoting aggregation of endogenous tau [27,28,29,30]. In this context, Shrivastava et al. studied the early events that occur on cell surfaces immediately before the endocytosis of tau fibrils. They demonstrated that fibrillar tau formed clusters at the plasma membrane of excitatory synapses following lateral diffusion, and that these clusters interacted with different membrane proteins, including Na+/K+-ATPase (NKA) and AMPA receptors. This interaction altered synaptic protein composition, reducing the amount of α3-NKA and increasing the amount of AMPA receptor subunit GluA2, the former reducing neuronal control of membrane depolarisation, and the latter making neurons more vulnerable following action-potential-induced Ca^2+^ influx. Overall, this study reported another mechanism through which pathological tau could affect synapse functionality, albeit in an advanced phase of disease, and potentially indicated new interactions as targets for decreasing tau fibril toxicity.

#### 2.2.2. In Vivo Models

Tau and Aβ have both been associated with memory impairment, mild cognitive impairment (MCI), and early AD [31,32,33]. However, it has not yet been established whether, and how, they might interact.

Different in vivo studies have tried to address this. It is well known that soluble Aβ oligomers exert an inhibitory effect on hippocampal LTP [34,35,36,37]. Shipton and colleagues [38] demonstrated that tau is required for this specific toxic effect of Aβ oligomers. Indeed, in tau KO mice, the absence of tau was sufficient for prevention of Aβ-induced LTP impairment. They also observed that tau became hyper-phosphorylated in the presence of Aβ oligomers, and that the specific inactivation of GSK3 inhibited not only Aβ-mediated tau phosphorylation, but also Aβ-mediated LTP deficits. These data support the hypothesis that tau, and specifically its phosphorylation by GSK3, are necessary for the toxic effects of Aβ on synaptic plasticity. It is possible that preventing tau and Aβ crosstalk may represent an innovative strategy for treating cognitive impairment.

**Figure 2 biomedicines-10-01550-f002:**
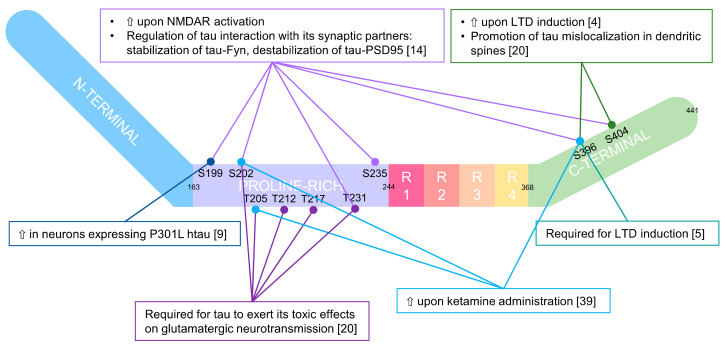
Graphic representation of main human tau phosphosites and indications of the molecular events regulating their phosphorylation. Tau is composed of two main domains: a projection domain (resides Met1-Tyr197) and an assembly domain (Ser198-Leu441). The first includes the N-terminal extension and the first part of a proline-rich region (residues 165–197), and the latter the second part of the proline-rich region, three or four microtubule binding repeats (MTBR, numbered R1 to R4) and a C-terminal extension [4,5,9,14,20,39].

Ittner and colleagues [11] demonstrated an alternative pathway through which tau could mediate Aβ toxicity in AD. As described above, they identified a physiological function for tau in the post-synaptic compartment where it recruits Fyn and promotes its kinase activity on GluN2B. In tau KO mice crossed with Aβ-forming APP23 mice, they observed a decrease in Fyn-mediated GluN2B phosphorylation and a reduction in memory deficits, as well as significantly delayed onset of mortality and an improved overall survival, without changing Aβ levels or plaque load. The positive effect on survival gained from the absence of tau may have resulted from a reduced sensibility to excitotoxicity due to reduced stabilisation of NMDAR in the postsynaptic density complex. Of note, they observed that a peptide that uncouples the Fyn-mediated interaction of NMDARs and PSD95 prevented premature lethality and memory deficits in Aβ-forming APP23 mice. These data indicated that Fyn-related dendritic function of tau confers Aβ toxicity at the post-synapse. These studies are consistent with the data presented by Liao’s research group which demonstrated in vitro that tau was necessary for observing Aβ oligomer induced AMPAR signalling deficits [19].

Monteiro-Fernandes and colleagues proposed an intriguing therapeutic strategy to counteract tau-mediated excitotoxic synaptic and memory deficits in Aβ-dependent hippocampal pathology [39]. They chronically administered a positive allosteric modulator (PAM) of AMPAR to a non-transgenic mouse model of Aβ-oligomers. They found that PAM treatment reverted Aβ-driven memory deficits and synaptic missorting of tau and the associated Fyn-GluN2B driven excitotoxic synaptic signalling. In this way, they demonstrated that acting on glutamatergic signalling, and specifically potentiating AMPAR, could prevent tau-mediated toxic synaptic effects, therefore providing a potential pathway for new and alternative treatments of AD and other tauopathies. 

Focusing on other aspects of tau, Li and colleagues [40] showed another mechanism through which it mediates neurotoxicity by affecting synaptic functioning. They demonstrated that long-term ketamine administration to WT mice caused excessive tau protein phosphorylation at Ser202/Thr205 and Ser396 in mouse hippocampus, with a decrease in hippocampal synaptic function and the number of AMPAR in the membrane. Interestingly, these last two effects were tau-dependent as they were not detectable in tau KO mice. These data are consistent with other studies that have indicated the involvement of tau in mediating the neurotoxic effects of various factors, such as anaesthesia and drugs [41,42]. Additionally, as discussed in Section 3.3.2, it seems that many different toxic factors converge on tau to mediate their effects on the synaptic compartment.

Overall, both in vitro and in vivo studies have demonstrated that tau, and specifically its phosphorylation, when dysregulated, negatively affects synapse functionality and glutamatergic signalling. In this light, a dysregulation of tau phosphorylation, and the subsequent alteration of its physiological function and localisation in spines, could be the very initial stage of tauopathies, preceding the misfolding of tau and the accumulation of toxic aggregates. Should this be the case, targeting of this very first event of tau dysregulation could represent a valid strategy for counteracting tauopathy disease progression.

## 3. α-Synuclein

### 3.1. α-Synuclein Conformations

First discovered in 1988, α-synuclein is a small, soluble intracellular protein of 15 kDa belonging to the synuclein superfamily, which also comprises β- and γ-synuclein [43]. α-Synuclein can be divided into three main domains with distinct physico-chemical properties (see Figure 3). The N-terminal region is characterised by seven 11-mer repeats with a KTKGEV consensus sequence (residues 1–95) [44]. This domain forms an amphipathic α-helix which confer to the protein the ability to bind negatively charged lipid membranes [45,46]. Aminoacids from 61 to 95 constitute the NAC (Non-amyloid β component) domain, thought to favour α-synuclein aggregation [47,48]. α-Synuclein C-terminus (aa 96–140) is highly acidic and mainly unstructured. It is targeted by post-translational modifications at various sites, including phosphorylation at Ser129, known to promote toxic species formation [49,50,51]. The C-terminal domain is responsible also for interaction with proteins and ions, modulation of membrane binding, and protection from protein aggregation [52,53,54]. α-Synuclein can mainly be referred to as a natively unfolded protein, whose structure changes depending on localisation, interaction, and membrane-binding status [55,56,57]. This protein exists in the cell as a dynamic pool of monomers and multimers in an equilibrium between soluble and membrane bound states due to its high affinity to anionic membranes [58,59]. 

Due to its structure and the presence of a hydrophobic non-amyloid β-component (NAC) region, α-synuclein has a high tendency to change its conformation into β-sheet rich structures that are prone to aggregating into oligomers, and insoluble amyloid fibrils that ultimately deposit in Levy Bodies (LB) inclusions [60]. Despite its ubiquitous expression across the nervous system, the pathological aggregation of α-synuclein and consequent neurotoxicity seem to selectively affect specific neuronal subpopulations [61]. 

The characterisation of the α-synuclein oligomer role in cellular damage and amyloid formation has been quite challenging because of their transient nature and high conformational variability. Oligomers can be described as a continuum of species going from low-molecular weight prefibrillar oligomers to larger aggregates that are progressing towards amyloid formation [62]. As demonstrated in recent studies, oligomeric species are emerging as having a leading role in causing neurotoxicity, including small and soluble forms [63,64,65].

The process of α-synuclein amyloid fibril formation is not completely understood. However, an alteration in the dynamic pool of structurally diverse α-synuclein forms, and defects in proteostasis mechanisms, may be relevant elements in this process [66,67]. A better comprehension of fibrillar structures has been provided mainly through NMR studies and, more recently, advancements in cryo-EM technology [68,69,70,71]. Interestingly, recent works postulate the existence of different strains of amyloid fibrils, capable of provoking distinct disease phenotypes [72]. In vitro, by varying the buffer and salt conditions, it has been possible to obtain diverse α-synuclein species as classic fibrils or ribbons, which feature flat and twisted structures. Upon injection into rodents, these ribbons seemed to produce α-synuclein inclusions, particularly in oligodendrocytes, which is a typical neuropathological sign of multiple system atrophy (MSA). Contrastingly, fibrils are reported to cause mainly PD-like features with loss of dopaminergic nigral neurons and motor impairments [68,73].

### 3.2. α-Synuclein Synaptic Functions

Highly expressed in neurons of the central and peripheral nervous system, α-synuclein has important neuronal functions at the presynaptic terminal, where it is predominantly expressed [74,75]. Indeed, studies on mice with all synuclein members knocked out described primarily pre-synaptic alterations, including reduced size of the synaptic terminal [76]. Almost no evidence of relevant postsynaptic activities of the endogenous protein have been demonstrated so far. The intrinsically unfolded nature of α-synuclein, its tendency to aggregate, and the possible presence of a compensatory effect from the β- and γ-synucleins make the study of its physiological functions particularly challenging. 

In more than twenty years of research, a multitude of cellular and synaptic functions have been attributed to α-synuclein, including dopamine and monoamine metabolism [77,78]. α-Synuclein interacts with multiple presynaptic partners and shows high affinity to curved anionic membranes, strongly supporting its modulatory function on the synaptic vesicle pool and plasticity events [47,77,79,80]. Indeed, α-synuclein colocalises and interacts with synaptic vesicles, synaptobrevin-2 and the G-protein Rab3a, and acts as a chaperone for the SNARE-complex assembly [43,59,81,82]. Contrasting results, however, have been reported regarding α-synuclein inhibition or promotion of neurotransmission [83]. Altogether, literature data indicate a role in intense and regulated neuronal activity rather than in basal neurotransmission [77]. In this regard, a recent study identified α-synuclein involvement in regulation of short- and long-term dopaminergic plasticity in vivo, particularly in the facilitation and depression of dopamine release [84].

### 3.3. Role of α-Synuclein Aggregates in Functional Alterations of Glutamatergic Synapses

Aberrant levels and forms of α-synuclein exert neurotoxicity through multiple mechanisms, affecting homeostatic cell pathways and synaptic functions. Several works describe key contributions of toxic forms of α-synuclein in perturbing synapse structure and activity. Amyloid and oligomeric forms of the protein were shown to spread between neurons transynaptically, making toxic forms of the protein detrimental to multiple aspects of synapse structure and neuronal transmission [85]. However, the broad spectrum of α-synuclein toxic aggregates that can be generated in vitro, and can be found in patients’ brains, complicates the precise investigation of molecular events. Interestingly, one hypothesis attributes clinical heterogeneity and different pathophysiological mechanisms of synucleinopathies to different strains of α-synuclein bearing distinct ultrastructural features [73]. Recent experimental evidence obtained by using both in vivo and in vitro models of synucleinopathies reported an early impact of α-synuclein on glutamatergic neurotransmission. Specifically, aberrant α-synuclein impacted the subunit composition and function of both NMDARs and AMPARs (see Table 1). However, α-synuclein-mediated toxicity at the glutamatergic post-synaptic compartment was strictly dependent on the structural biophysical characteristics of the protein aggregates and of the neuronal subtype being considered.

#### 3.3.1. In Vitro Models

##### Effect of α-Synuclein Monomers

One of the mechanisms responsible for α-synuclein misfolding and aggregation is the increased expression of the protein and, consequently, the augmented intra- and extracellular concentration of α-synuclein monomers. Indeed, overexpression of the protein or exposure to increased exogenous monomer concentration in in vitro systems have been reported to affect the glutamatergic synapse. In this regard, Chen et al. demonstrated that both α-synuclein overexpression, and the exogenous treatment of primary hippocampal neurons with α-synuclein monomers, significantly decreased membrane levels of NMDARs [86]. In particular, aberrant internalisation of the NMDAR complex was indicated by decreased surface levels of the obligatory NMDAR subunit GluN1 concomitant to increased cytoplasmic expression. As well as promoting NMDAR endocytosis, α-synuclein reduced receptor functionality, as shown by impaired NMDAR-mediated calcium influx. Interestingly, NMDAR defects were rescued by downregulation of Rab5a, suggesting that it acts as a mediator of α-synuclein-induced NMDAR endocytosis. In agreement with this, whole-cell patch clamp recordings of treated neurons revealed a marked reduction in NMDAR-elicited inward currents [86]. Navarria and colleagues [90] showed that the physiological expression of wt-α-synuclein or a c-terminally truncated form negatively affected NMDAR surface levels. A significant increase of GluN2B and GluN1 expression upon genetic deletion of α-synuclein was found in primary cortical neurons, obtained from α-synuclein-null mice, when compared with wt-mice-derived neurons. Moreover, overexpression of truncated α-synuclein in SK-N-SH cells led to increased NMDAR subunit expression and cytoplasmic localisation. Pre-treatment of cells with the GluN2B selective inhibitor ifenprodil abrogated the NMDAR-calcium elevation, indicating a specific effect of α-synuclein on the GluN2B subunit [90]. 

Another study investigated the effects of aberrant extracellular concentrations of monomeric α-synuclein on neuronal lipid raft composition and postsynaptic activity [91]. The integrity and molecular composition of these cholesterol-enriched membrane domains are fundamental for the maintenance of proper pre- and post-synaptic structure and, consequently, receptor function [98]. Emanuele and colleagues characterised an indirect effect of increased extracellular α-synuclein that, through fragmentation of lipid rafts, drove important remodulation of the pre- and postsynaptic architecture [91]. Despite not changing the total levels of the GluN2B subunit and PSD95, α-synuclein exposure modified lipid raft partitioning of these two postsynaptic proteins in corticostriatal slices. PSD95/GluN2B interaction was also significantly reduced, possibly due to augmented GluN2B phosphorylation at Ser1480, which was already known to modulate its interaction with PSD95 [99]. Along with these modifications, α-synuclein elicited functional impairments, blocking the induction of chemical LTP. Specifically, α-synuclein dampened NMDARs responses in presence of normal AMPAR mediated mEPSCs [91]. 

##### Effect of Oligomers and Fibrils

A possible explanation for the results obtained with the use of α-synuclein monomers could reside in the unstable nature of α-synuclein and its ability to rapidly form aggregates that can contribute to the described neurotoxicity. Therefore, the consequences of direct application of toxic α-synuclein aggregates have been subsequently investigated in more detail. As described above, oligomeric and fibrillar species can comprise a wide spectrum of dimensionally diverse aggregates formed from various proportions of soluble and insoluble amyloid forms.

The described capability of α-synuclein monomers to induce endocytosis of the NMDAR complex was further described by a study published in 2019. In particular, this study correlated this process to different preparations of α-synuclein oligomers [87]. Firstly, the authors showed that direct application of oligomers to MES23.5 dopaminergic cells led to an increased ability to promote GluN1 subunit internalisation with respect to the monomeric form. Interestingly, aggregates generated by incubating α-synuclein in the plasma of PD patients provoked the most significant reduction of surface GluN1. Importantly, treatment with a clathrin inhibitor was able to rescue oligomer-induced NMDAR internalisation, suggesting clathrin-mediated endocytosis could be an affected pathway when challenged by α-synuclein. These species showed increased size and content of phosphorylated α-synuclein when compared with oligomers aggregated in the plasma of control individuals. In this regard, an earlier work using a similar in vitro model, consisting of autaptic cultures of hippocampal primary neurons, explored how biophysically diverse oligomers can affect excitatory neurotransmission [97]. Incubation of monomeric proteins with solvents (EtOH and DMSO) in combination with Fe^3+^ ions allowed stabilisation of oligomers to a higher molecular order. This study did not report any effect on NMDAR function on autaptic neurons upon application of nanomolar iron-induced oligomers, but patch clamp recordings revealed a significant increase in the amplitude of AMPAR-mediated EPSCs. Neither smaller aggregates nor monomeric proteins were able to elicit similar AMPARs alterations in this experimental setting. Therefore, authors speculated that the molecular mechanism behind these alterations likely involved an increased responsiveness, or quantity, of AMPARs, without affecting the total number of synapses. Indeed, the demonstrated detrimental effects of large oligomers on membrane integrity could also be involved in these alterations. In particular, the formation of calcium-permeable pores could further increase glutamate-induced excitotoxicity, contributing to neuronal death [97]. 

In hippocampal slices, high-order oligomeric species were demonstrated to impair induction of synaptic plasticity through NMDARs-dependent mechanisms [96]. Incubation of large oligomers on rodent hippocampal slices blocked LTP induced by theta-burst stimulation. The same effect was elicited by oligomers stabilised by 4-hydroxy-2-nonenal (HNE), a lipid peroxidation product, with aggregates bearing post-translational modifications comparable to those found in PD-brains. Notably, the same concentration of monomeric or fibrillar α-synuclein did not reduce LTP, suggesting this was a specific effect for oligomeric species. Although LTP defects were demonstrated to be strictly dependent on the activation of NMDARs, oligomers triggered downstream alterations to AMPARs as well. Specifically, oligomer-treated slices showed a significant increase in current rectification at positive potentials as a consequence of AMPARs lacking GluA2. Pathological changes in AMPAR function following NMDAR overactivation was further supported by an increased surface localisation of the GluA1 subunit. The authors suggested that these results indicated a pathological mechanism involving aberrant Ca^2+^ influx downstream of NMDAR and AMPAR alterations. Indeed, aberrant calcium elevation can lead to increased activation of calcium dependent phosphatases such as calcineurin, which has already been described as participating in neurotoxic events [100].

In parallel to the characterised postsynaptic toxicity in the hippocampus, oligomeric forms of α-synuclein can also affect corticostriatal glutamatergic signaling, altering the postsynaptic activity of NMDARs [88]. Proper composition and localisation of these receptors is physiopathologically important; deregulation of the GluN2A/GluN2B subunit ratio at striatal spiny projection neurons (SPNs) was demonstrated to contribute to motor behaviour and synaptic plasticity impairments in PD [101]. Durante and colleagues [88] investigated the impact of nanomolar concentrations of small soluble α-synuclein oligomeric species on corticostriatal plasticity. In particular, α-synuclein prevented the induction of LTP, mediated primarily by NMDAR activation, at the level SPNs, but retaining the capability for LTD events. Interestingly, the authors found that AMPAR-mediated synaptic currents and AMPAR rectification index were not affected by oligomer exposure, while a selective decrease in GluN2A-mediated currents was recorded. In line with functional findings, the synaptic expression of AMPAR remained unchanged, in contrast to a selective reduction of the GluN2A-subunit [88].

While the postsynaptic impact of α-synuclein oligomers towards glutamate receptors has been largely addressed, knowledge about fibrillar species still remains mostly unknown and is made more complex by the extreme difference in the biophysical properties of amyloid aggregates. Although fibrils represent the most utilised species in generating in vivo models of synucleinopathies, only one recent study has addressed the molecular events triggered by diverse α-synuclein fibrillar polymorphs, with each one connected to a specific synucleinopathy [92]. The authors showed that structurally diverse α-synuclein fibrils (fibrils and fibrils-91) and ribbons bind to membranes and excitatory synapses of both hippocampal primary neurons and organotypic cultures, with varying efficiency. Additionally, they found that distinct polymorphs triggered specific alterations; fibrils selectively increased the clustering at synapses of GluA2 and GluN2B subunits, fibrils-91 specifically targeted GluA2, while ribbons did not affect any of the evaluated glutamate receptor subunits. Therefore, α-synuclein-driven redistribution of excitatory postsynaptic proteins seems strictly related to polymorph structural features, underlying the different possible alterations of homeostatic pathways. Mislocalisation of glutamate receptors induced by fibrillar polymorphs was reflected in significant impairments to neuronal network activity, further supporting the detected upstream defects in glutamatergic synapse composition [92].

#### 3.3.2. In Vivo Studies

The recent characterisation and validation of various rodent models of synucleinopathies has unveiled the different mechanisms of toxicity through which α-synuclein affects glutamatergic transmission in vivo. In 2016, Tozzi et al. identified a specific pathogenic mechanism mediated by GluN2D-containing NMDARs in striatal cholinergic interneurons, underlying early behavioural and electrophysiological defects in mice and rats overexpressing α-synuclein. While SPNs showed normal electrophysiological properties, LTP induction was selectively abolished in cholinergic interneurons. Specifically, α-synuclein-overexpression induced a selective modulation of GluN2D-containing NMDARs, contributing to impairment of plasticity as well as to early memory and motor alterations [94]. Notably, a subchronic treatment with L-DOPA rescued these cholinergic defects; it can therefore be speculated that partial dopamine deficiency has a role, together with the α-synuclein-induced modulation of NMDARs, in these pathophysiological events.

As discussed above, the detrimental effects of oligomeric α-synuclein on NMDAR-dependent synaptic activity has already been established ex vivo on corticostriatal plasticity. The in vitro α-synuclein treatment of corticostriatal slices, which reduced NMDAR currents and impaired the induction of LTP in SPNs, selectively reducing the postsynaptic localisation of the GluN2A subunit was also discussed [88]. In line with this finding, the in vivo striatal injection of different α-synuclein toxic species in rats caused visuospatial alterations that were strictly dependent on NMDAR activation in the dorsal striatum [88]. Data from this study suggested that protofibrils (PFFs) and oligomers of α-synuclein mediated NMDAR-dependent defects to different extents. Specifically, animals inoculated with oligomers displayed more severe alterations to spatial and novel object recognition abilities when compared with rats injected with PFFs. Taken together, these studies indicate that specific concentrations and structural features of α-synuclein aggregates can differentially impact selective neuronal populations and NMDAR subunits in vivo.

A more recent study by the same research group deepened the understanding of detrimental in vivo effects of α-synuclein towards the nigrostriatal circuitry, focusing on the role of PFFs in the striatum [89]. Prior to significant α-synuclein deposition, PFF-lesioned rats showed specific behavioural alterations accompanied by electrophysiological impairments to dopaminergic SNpc neurons. Interestingly, striatal LTP of SPNs was strictly dependent on NMDAR activity and was significantly impaired by 6 weeks post PFF injection. Conversely, defects in striatal LTD, a form of plasticity more dependent on dopamine transmission, started to appear 12 weeks after α-synuclein seeding. Although a subchronic treatment with L-DOPA restored both forms of synaptic plasticity, the recovery of LTP was only partial. Interestingly, these findings suggested that the block of LTP induction in SPNs can therefore primarily be ascribed to α-synuclein impact on postsynaptic NMDARs. 

Using a combination of in vivo and ex vivo studies, Ferreira et al. described another possible deregulated pathway contributing to α-synuclein-induced NMDAR toxicity, mediated by the GluN2B subunit [93]. Oligomeric forms of the protein impaired LTP ex vivo in both hippocampal and striatal rodent slices [88,96]. Here, the authors described a possible interplay between α-synuclein, the cellular prion protein (PrPC), Fyn kinase, and GluN2B-containing NMDARs in the hippocampus. In particular, genetic deletion of PrPC reversed the LTP impairments induced by application of oligomeric species to hippocampal slices. Similarly, inhibition of Fyn kinase, which is involved in NMDAR phosphorylation and consequent excitotoxicity, also rescued these functional synaptic impairments. Using cultured primary neurons, authors subsequently showed that PrPC is needed for α-synuclein-mediated activation of Fyn. Consequent aberrant Fyn-mediated GluN2B phosphorylation could therefore contribute to the molecular events underlying LTP blockade. Additionally, mGluR5 was identified as an important player in physiopathological GluN2B-Y1472 phosphorylation. Further supporting these results, blockade of this pathway in vivo reversed hippocampal-dependent memory deficits in α-synuclein-transgenic mice.

Combining both in vitro and in vivo studies, a recent paper described possible dual pathophysiological mechanisms through which a mixture of soluble α-synuclein aggregates modulate glutamatergic transmission [95]. α-synuclein preparations in vitro induced abnormal local glutamate release from cultured astrocytes, a process that was then confirmed in vivo. Glutamate microdialysis measurements in transgenic α-synuclein-mice showed elevated tonic glutamate release. Moreover, the increased basal inward currents detected in the CA1 region of hippocampal slices indicated a subsequent aberrant activation of neuronal glutamate receptors, including extrasynaptic NMDARs. Parallel to this indirect receptor activation, oligomeric α-synuclein was shown to directly activate NMDARs on slices exposed to the α-synuclein aggregate preparation. The hypothesised aberrant activation of mainly extrasynaptic NMDARs was further supported by the rescue of α-synuclein-dependent synaptic damage by application of a selective antagonist of extrasynaptic NMDARs.

## 4. α-Synuclein and Tau: Additional Molecular Mechanisms

Even if we focused above on the toxic effects exerted by tau and α-synuclein on AMPAR and NMDAR-mediated glutamate neurotransmission, it is relevant to mention that glutamate can indirectly trigger tau aberrant phosphorylation and contribute to α-synuclein misfolding by other key molecular mechanisms. Indeed, different in vitro and in vivo studies demonstrated that glutamate mediated excitotoxicity is one of the factors which supports ER stress [102,103,104,105]. Relevantly, it is well known that ER stress promotes toxic GSK-3β activation and, consequently, hyper-phosphorylation of tau [106,107,108]. In this view, we can hypothesise that altered glutamate neurotransmission, affecting ER activity, could promote tau aberrant phosphorylation and subsequent toxic effects. In other words, it could be that, under certain circumstances, the toxic crosstalk between tau and glutamate ionotropic receptors goes also in the opposite direction to what we have described until now. Moving to α-synuclein, ER stress has been described as one of the mechanisms that promotes α-synuclein misfolding and aggregation [109,110]. Interestingly, aberrant α-synuclein species trigger glutamate neurotoxicity (see Section 3.3.1) directly acting on the synapse or activating inflammatory pathways as discussed above. Therefore, in the context of synucleinopathies it can be speculated that aberrant glutamate response can also contribute to α-synuclein aggregation in a pathogenic positive feed-forward loop.

In addition, ER stress possibly contributes to proteinopathies pathogenesis also through dysfunction of ER chaperons Sigma receptors (Sig-R). Indeed, emerging evidence pointed out a role for Sig-Rs in the modulation of tau phosphorylation. Tsai and colleagues [111] demonstrated that Sig1-R promotes the degradation of the tau kinase CDK5. In addition, the administration of Sig1-R agonist inhibits tau protein phosphorylation by reducing the activity of GSK-3β [112,113]. These discoveries suggest that Sig-Rs act as negative regulators of the two main kinases of tau and, thus, they could have a role in tauopathies and represent new molecular targets. In the context of synucleinopathies, Hong and colleagues reported increased levels of α-synuclein phosphorylation and aggregation in mice knockout for Sig1-R, eventually leading to motor defects and increased dopaminergic degeneration [114]. Another recent work supported the relevance of Sig-Rs in the process of α-synuclein pathogenic misfolding. Indeed, compounds blocking Sig2-R activity were found effective in counteracting detrimental effect of syn oligomers on lipid vesicle trafficking and lysosomal functions in neurons [115].

## 5. Conclusions

As described above, several studies in the last decade have clearly demonstrated complex and diversified crosstalk of tau and α-synuclein with ionotropic glutamate receptors localised at excitatory postsynaptic sites. While it has been shown that tau also has physiological roles in dendritic spines modulating synaptic plasticity and receptor function, α-synuclein activity at postsynaptic sites seems restricted to the induction of toxic events associated with receptor endocytosis and loss of synaptic plasticity. However, aberrant protein mislocalisation at postsynaptic sites represents a well-demonstrated and common event for both hyperphosphorylated tau and α-synuclein aggregates and plays a key role in driving postsynaptic toxicity and induction of early neurodegenerative mechanisms. 

Studies here reported clearly indicate that both physiological and pathological functions for tau and α-synuclein must be considered. Accordingly, deepening our knowledge of specific synaptic functions could also help the development of more precise and appropriate therapies for all those pathological and neurodegenerative conditions in which their correct function is lost. This issue is highly relevant since no pharmacological strategies have been described that are capable of directly intervening on the action of tau and synuclein at the level of the ionotropic glutamate receptors. It is hopefully possible that further knowledge of the molecular mechanisms through which tau and synuclein act physiologically and pathologically on NMDARs and AMPARs will aid researchers in the identification and preclinical evaluation of new molecules capable of restoring the functionality of glutamatergic transmission in the early stages of proteinopathies.

## Figures and Tables

**Figure 1 biomedicines-10-01550-f001:**
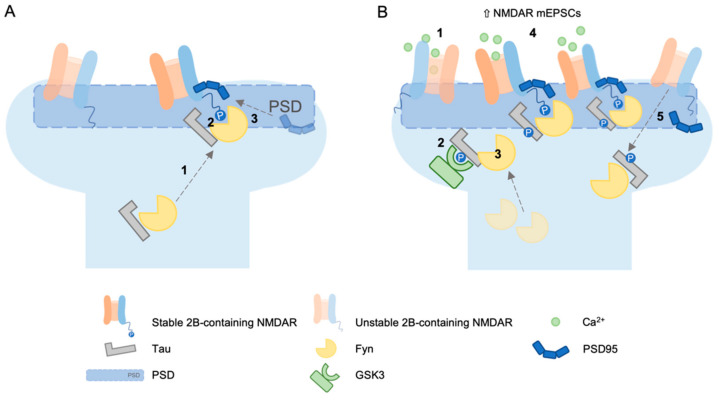
(**A**) Fyn localises to the postsynapse in a tau-dependent manner (1) where it associates with the postsynaptic density and phosphorylates GluN2B in the carboxy terminus domain (2). Since this phosphorylation facilitates the interaction of NMDARs with the scaffolding protein PSD-95 (3), Tau–Fyn interaction in spine increases the stability of NMDAR within the PSD [11]. (**B**) According to the Mondragón-Rodríguez model [12], the activation of NMDAR (1) leads to phosphorylation of tau by GSK3 (2). This phosphorylation strengthens the interaction between tau and Fyn (3), thus supporting a transient increase in NMDA receptor activation (4), but weakens the interaction between PSD95 and tau. As a result, Fyn finally leaves the PSD95-NMDA receptor complex (5).

**Figure 3 biomedicines-10-01550-f003:**
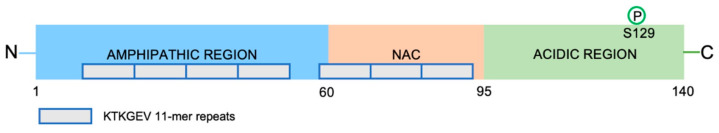
Schematic representation of the secondary structure of the protein αsyn.

**Table 1 biomedicines-10-01550-t001:** Table reporting impact of different α-synuclein species on specific ionotropic glutamate receptor (iGluR) subunits.

Targeted iGluR	α-Synuclein Species	Model	Mechanism	Ref.
GluN1	Overexpression, monomers	Primary hippocampal neurons	⇧ internalisation, impaired calcium influx	[86]
	Oligomers	MES23.5 dopaminergic cells	⇧ internalisation	[87]
GluN2A	Small oligomers	Corticostriatal slices	LTP blockade in SPNs, ⇩ GluN2A currents, ⇩ GluN2A expression	[88]
	Protofibrils	Striatal injection (rats)	LTP, LTD impairments (SPNs), behavioral alterations	[89]
GluN2B	Overexpression (wt/αsyn1-120)	SK-N-SH, primary cortical neurons, Tg-αsyn1-120-mice	⇧ internalisation, impaired calcium influx	[90]
	Monomers	Corticostriatal slices	Mislocalisation, LTP blockade	[91]
	Fibrils	Primary hippocampal neurons hippocampal organotypic slices	⇧ synaptic expression, network activity impairment	[92]
	Oligomers	Hippocampal slices primary hippocampal neurons tg-αsyn-mice	LTP impairments, PrPC/Fyn/GluN2B deregulation	[93]
GluN2D	In vivo overexpression	Tg-αsyn1-120-mice AAV-αsyn-rats	LTP-blockade (ChIs), Memory and motor alterations	[94]
Extra-synaptic NMDARs	Soluble aggregates (oligomers+fibrils)	Primary astrocytes tg-αsyn-mice cerebrocortical cultures hippocampal slices	Astrocytes-mediated glutamate release, direct activation of extra-synaptic NMDARs	[95]
GluA1	HNE-stabilised oligomers	Hippocampal slices	⇧ AMPAR rectification currents, impaired LTP	[96]
GluA2	Fibrils-91	Primary hippocampal neurons hippocampal organotypic slices	⇧ synaptic expression, network activity impairment	[92]
AMPARs	Iron-induced oligomers	Autaptic cultures of primary neurons	⇧ AMPAR-mEPSCs ⇧ responsiveness/AMPAR n°	[97]

## Data Availability

Not applicable.

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
