# Peer review of "NMDA and AMPA Receptors at Synapses: Novel Targets for Tau and α-Synuclein Proteinopathies"

_biomedicines, 2022, doi:10.3390/biomedicines10071550_

Round 1

Reviewer 1 Report

Review article titled (NMDA and AMPA receptors at synapses: novel targets for tau and alpha-synuclein proteinopathies) by Italia et al. discussed the improtance of 2 important typesof glutamate receptors, NMDA & AMPA, at the synaptic level and theri importance as therapeutic targets for tau and alpha synuclein proteinopathies.

1- In Tau & alpha-synuclecin : separate the in vivo form ex vivo studies the same way to be symmetric & put a subtitle for each one of them. Hence sections become shorter and easy for readers.

2- It is better not to abbreviate alpha=synuclein to syn, consider this in subtitles & text.

3- Use PD & AD: for Parkinson's disease and Alzheimer's disease

4- I wish if the authors write 2 sentences about future directions after this review.

5- Add the chemical structure of NMDA & AMPA

6- Add a scheme for the protein structure of alpha  synuclein & tau.

Author Response

Review article titled (NMDA and AMPA receptors at synapses: novel targets for tau and alpha-synuclein proteinopathies) by Italia et al. discussed the improtance of 2 important types of glutamate receptors, NMDA & AMPA, at the synaptic level and their importance as therapeutic targets for tau and alpha synuclein proteinopathies.

1- In Tau & alpha-synuclein: separate the in vivo form ex vivo studies the same way to be symmetric & put a subtitle for each one of them. Hence sections become shorter and easy for readers.

Reply: We want to thank the Reviewer for this suggestion that we have carefully taken into consideration. In the revised version of the review, we have rearranged these sections. In particular, we have divided tau and alpha-synuclein sections in the same subsections, namely “in vitro models” and “in vivo models”. However, we did not divide ex-vivo from in-vivo studies because almost all studies performed by using in vivo models that we mentioned include both in vivo behavioral observations and ex-vivo molecular/electrophysiological data.

2- It is better not to abbreviate alpha-synuclein to syn, consider this in subtitles & text.

Reply: We agree on this point and the text has been modified accordingly.

3- Use PD & AD: for Parkinson's disease and Alzheimer's disease

Reply: We agree on this point and the text has been modified accordingly.

4- I wish if the authors write 2 sentences about future directions after this review.

Reply: As requested by the Reviewer we have modified the Conclusions section of the review including additional sentences related to future directions that should be taken in this field.

5- Add the chemical structure of NMDA & AMPA

Reply: We agree on this point and the text has been modified accordingly.

6- Add a scheme for the protein structure of alpha-synuclein & tau.

Reply: We want to thank the Reviewer for this suggestion. In the revised version of the review we have added a new figure (see Figure 3) including alpha-synuclein protein structure and we have modified Figure 2 containing now a fully revised and more appropriate figure legend.

Reviewer 2 Report

Interesting paper looking at the role of NMDA and AMPA receptors for proteinopathies and neurodegenerative disease. The authors do a good job looking at the receptors and phosphorylation states. 

Authors should focus more specifically on the glutamate response and how excitotoxicity can activate the ER stress cascade to contribute to protein aggregation. 

Some key references are lacking and should be included to discuss modulation of sigma receptors PMID: 27580401.

If this is appropriately addressed and above references included, could be of interest to the readership. 

Author Response

Interesting paper looking at the role of NMDA and AMPA receptors for proteinopathies and neurodegenerative disease. The authors do a good job looking at the receptors and phosphorylation states. Authors should focus more specifically on the glutamate response and how excitotoxicity can activate the ER stress cascade to contribute to protein aggregation. 

Some key references are lacking and should be included to discuss modulation of sigma receptors PMID: 27580401.

If this is appropriately addressed and above references included, could be of interest to the readership. 

Reply: The Reviewer raises an interesting point. We need to mention that our review is mostly addressing the role of synaptic enriched glutamate receptors as novel targets for tau and alpha-synuclein proteinopathies considering that it has been prepared for the special issue “News about Structure and Function of Synapses: Health and Diseases”. However, we appreciated the suggestion of this Reviewer and in the revised version of the review we have now added a fully new section describing the role of ER stress and sigma receptors.

Reviewer 3 Report

Here I present my comments on the review entitled NMDA and AMPA receptors at synapses: novel targets for tau and alpha-synuclein proteinopathies“ presented by Italia et al. for publication in Biomedicines (Manuscript ID: biomedicines-1766334)

 This Review is very is well written and presented in a comprehensive manner. Also the figure and table are displayed clearly. In general, I felt enthusiastic while I was reading the text and I enjoyed the way the authors separate the different subtopics. For example, they nicely constructed from the classical functions and views of Tau in  microtubule stabilization and axonal elongation in the somato-axonal compartment and Tau phosphorylation to newer and provocative ideas.

 One detail. I would love to see the introduction of a cartoon displaying the Author‘s propositions on the PSD-Fyn-NMDAR complex and regulatory role of GSK3) β-mediated phosphorylation of tau and internalization of AMPA receptors. This is some extra-work, therefore, I would leave the decision on doing so to the Author‘s criteria.

 Beside this, I feel that this articule can be published in its current state.  

Author Response

Here I present my comments on the review entitled “NMDA and AMPA receptors at synapses: novel targets for tau and alpha-synuclein proteinopathies“ presented by Italia et al. for publication in Biomedicines (Manuscript ID: biomedicines-1766334)

This Review is very is well written and presented in a comprehensive manner. Also the figure and table are displayed clearly. In general, I felt enthusiastic while I was reading the text and I enjoyed the way the authors separate the different subtopics. For example, they nicely constructed from the classical functions and views of Tau in  microtubule stabilization and axonal elongation in the somato-axonal compartment and Tau phosphorylation to newer and provocative ideas.

One detail. I would love to see the introduction of a cartoon displaying the Author‘s propositions on the PSD-Fyn-NMDAR complex and regulatory role of GSK3) β-mediated phosphorylation of tau and internalization of AMPA receptors. This is some extra-work, therefore, I would leave the decision on doing so to the Author‘s criteria. Beside this, I feel that this articule can be published in its current state.  

Reply: We want to thank the Reviewer for his/her positive comments. Following Reviewer’s suggestion, we have added a new Figure (see Figure 1) related to the functional modifications of PSD95-Fyn-NMDAR complex.

Round 2

Reviewer 1 Report

Thanks

Reviewer 2 Report

Incorporate PMID: 27580401 in discussion of sigma receptor contribution. Once incorporated, paper is clear for acceptance.